# A Focus on Enterochromaffin Cells among the Enteroendocrine Cells: Localization, Morphology, and Role

**DOI:** 10.3390/ijms23073758

**Published:** 2022-03-29

**Authors:** Rita Rezzani, Caterina Franco, Lorenzo Franceschetti, Marzia Gianò, Gaia Favero

**Affiliations:** 1Anatomy and Physiopathology Division, Department of Clinical and Experimental Sciences, University of Brescia, 25123 Brescia, Italy; c.franco@unibs.it (C.F.); lorenzofranceschetti@gmail.com (L.F.); marzia.giano@gmail.com (M.G.); gaia.favero@unibs.it (G.F.); 2Interdipartimental University Center of Research “Adaption and Regeneration of Tissues and Organs—(ARTO)”, University of Brescia, 25123 Brescia, Italy; 3Italian Society of Orofacial Pain (SISDO), 25123 Brescia, Italy

**Keywords:** enteroendocrine cells, enterochromaffin cells, bipolarity, morphology, serotonin, melatonin, substance P

## Abstract

The intestinal epithelium plays a key role in managing the relationship with the environment, the internal and external inputs, and their changes. One percent of the gut epithelium is represented by the enteroendocrine cells. Among the enteroendocrine cells, a group of specific cells characterized by the presence of yellow granules, the enterochromaffin cells, has been identified. These granules contain many secretion products. Studies showed that these cells are involved in gastrointestinal inflammatory conditions and hyperalgesia; their number increases in these conditions both in affected and not-affected zones of the gut. Moreover, they are involved in the preservation and modulation of the intestinal function and motility, and they sense metabolic–nutritional alterations. Sometimes, they are confused or mixed with other enteroendocrine cells, and it is difficult to define their activity. However, it is known that they change their functions during diseases; they increased in number, but their involvement is related mainly to some secretion products (serotonin, melatonin, substance P). The mechanisms linked to these alterations are not well investigated. Herein, we provide an up-to-date highlight of the main findings about these cells, from their discovery to today. We emphasized their origin, morphology, and their link with diet to better evaluate their role for preventing or treating metabolic disorders considering that these diseases are currently a public health burden.

## 1. Introduction

The intestinal epithelium is one of the largest exposed surfaces of the human body and is a very important interface for integrating microenvironmental information with physiological signals from the immune, cardiovascular, and nervous systems [1,2,3]. It is generally associated with dietary nutrients; inflammatory agents and microbiota products modulate many biochemical pathways such as digestion, metabolism, pain, and immunity [2,4,5]. In the human gut, there are around 100 million intestinal endocrine cells, defined as enteroendocrine cells (EECs), found from the stomach to the rectum [5,6]. These cells are scattered, for example, along the epithelial cells in the crypts and villi of intestinal organs and represent 1% of the total gut epithelium [7,8].

In the year 1960, EECs were found to have markers for neuronal differentiation, and, for this reason, they were believed to originate from the neural crest and were named neuroendocrine cells [9]. Then, new technical approaches showed that EECs are derived from the endoderm and not the neuroectoderm, even if they share features with neurons [5,10]. They arise from pluripotent stem cells at the base of the gut crypts and migrate up the crypt–villus axis; stem cells differentiate into EECs over a period of 2–6 days thanks to three basic helix–loop–helix transcription factors as reported by Li et al., (2011) and Latorre et al., (2016). The same stem cells give rise to other epithelial cell types of the gut (enterocytes, Paneth, goblet cells), and these cells are present in the villi and crypts of the gut as shown in Figure 1. They are shed by an apoptotic process into the intestinal lumen at the end of their lifespan [3,10].

Since ECs are scattered along the epithelium and by far outnumbered by absorptive and enterocytes and goblet cells as reported above, they are very difficult to study. In the past many new and powerful genetic approaches for studying cells in mice, have been created; they allowed for a clear characterization of biochemical and cell biological properties of EECs [11]. In particular, the characterization of these cells becomes more crucial considering the increasingly central role of the gut-brain-microbiota axis and the mutual relationship that exist between the axis and the cells themselves. One of the main functions of the intestinal endocrine component is the production of hormones and signal molecules, such as serotonin (5-HT). The main point is that the physiological roles of gut 5-HT have more recently begun to move away from paracrine effects within the gut [12]. Studies reported that 5-HT has a role in bone formation and promotes lipolysis from adipocytes providing substrates for hepatic gluconeogenesis during fasting. Not only this, though, in fact, gut-derived 5-HT also reduces energy expenditure [13] and plays a key role in driving multiple physiological adaptations to nutrient deprivation [12]. At the same time, there are many conditions in which the presence of enteroendocrine cells (major producers of 5-HT in the gut) changes. For this reason, in this review we have tried to condense the information known to date about these cells and their activity, emphasizing their origin, plasticity, morphology, and physiology to better evaluate their physiological role and their potential therapeutic applications. We critically analyzed the literature which provides better insight into the mechanistic link of diet interactions and ECs in some metabolic diseases. The potential for diet interventions as a promising strategy for modulating the responses of EC to food ingestion, and then, preventing or treating metabolic diseases has been highlighted considering that these diseases are currently a public health burden.

## 2. Enteroendocrine Cells

Based on their cytoarchitecture and location, EECs are divided into open type, with a bottle-neck shape and a prolongation with microvilli in direct contact with the intestinal lumen, and closed type, strictly linked to the basal membrane, without reaching the intestinal lumen and without microvilli (Figure 2).

Open-type cells are activated by luminal content, while closed-type EECs are sensitive to luminal contents through neural or humoral networks. They accumulate the secretory products in cytoplasmic granules and release them at the basolateral membrane upon several chemical, neural, and other signals [5,8]. Based on the secretory products of EECs, Latorre and colleagues (2016) reported that many types of these cells exist; moreover, each EEC releases more than one signaling product [8]. 

The EECs and the main products in the gastrointestinal (GI) tract are shown in Figure 3 and summarized in Table 1.

### 2.1. Role of Enteroendocrine Cells in the Brain-Gut Axis

The bidirectional network between the brain and gut is an important research focus even if many questions are still unknown; there is no data about how this relationship changes during pathophysiological states. EECs share the same role that the gut has with brain in order to manage several mechanisms of GI tract physiology [15]. Vagal afferent fibers reach and innervate the wall of the GI tract; therefore, they are in contact with the intestinal epithelium. The products secreted by EECs act on receptors located along the vagal afferent fibers and activate them. The activated vagal afferent fibers interact with the brainstem (nucleus tractus solitarii and nodose ganglion) before reaching the brain. The end result is that the neuronal information regulates numerous functions such as motility, inflammatory response, immunity, and blood flow [16]. (Figure 4). Thus, EEC–brain connections could be a new tool to understand the role of EECs [8].

### 2.2. Enteroendocrine Cells and Plasticity

As reported above, intestinal stem cells generate absorptive enterocytes and all secretory cell types, including EECs, Paneth, and goblet cells (Figure 5).

EECs have a common progenitor; this common progenitor will create two other cell types called biased and non-biased progenitor enterochromaffin cells (ECs). Therefore, ECs derive from a separate EC-biased progenitor, while the other hormone-producing cells have a non-EC biased progenitor. This differentiation starts from the intestinal crypts and the cells stay in the same location for approximately 40–60 h. After this period, most of them move towards the intestinal villi and they start to produce and secret hormones/products as we can see in Figure 5. A small part of this population remains in the crypt for a longer time [18].

EECs cells are not present at a constant ratio across the GI tract, but surprisingly, all EECs show plasticity, and they can reacquire stem cell state upon damage and during chemotherapy or irradiation. So, they change state if needed [3,5,19].

Normally, EECs are distinguished by secretion products: ECs (serotonin, melatonin, substance P, and, in minor amount, secretin, gamma-aminobutyric acid, guanylin and sometimes cortico-releasing hormone); I cells (cholecystokinin), X cells (ghrelin), L cells (glucagon-like peptide), D cells (somatostatin), K cells gastric inhibitory peptide), I cells (cholecystokinin), and N cells (neurotensin) [20,21,22].

ECs are the most abundant EECs in the GI tract and are distributed widely in the gastric antrum, in the small and large intestine [23] (Figure 6). Giver their abundant distribution, we decided to summarize the discovery, localization, and the role of these cells in the GI tract.

## 3. History of Enterochromaffin Cells

In 1868, Heidenhain (1834–1897) was the first scientist to stain ECs in the gastric mucosa of a rabbit and a dog describing their yellow cytoplasmic granules [24]. In particular, the name was chosen because of their specificity to bind chromium–salt to form a colored deposit [25,26].

In 1897, the Russian scientist Kulchitsky (1856–1925) played an important role in defining these cells leading to the important identification of a diffuse neuroendocrine system in the gut. He evaluated granular cells in the GI crypts of cats and dogs. He reported his findings saying these words “In the epithelium coverage of the intestinal tract, I had the opportunity to study elements which, as far I know, have not been described by other scientists up to now and which are with no doubt of great interest in relation to the present knowledge of histology of the intestinal tract” (Figure 7) [27].

At the end of the staining processes, the granules took up fuchsin acid and became red with acidophil characteristics as previously described by Heidenhain (1870) [28]. In this period, there were few data about biochemical mechanisms and their functions could not be defined. This caused much controversy among researchers resulting in much confusion and misinterpretation. The same cells were described as cells of Nicolas-Kulchitschy, yellow cell of Schmidt, enterochromaffin cells of Ciacco, argentaffin or silver reducing cells of Masson and chromoargentaffin of Cordier based on their investigator [29]. In 1906, Ciacco responded to these controversies, and he suggested that the term ECs should have been used for defining the real characteristics of these cells and their anatomical location [30].

## 4. Ultrastructural Features of Enterochromaffin Cells and Their Bipolarity

The evaluation of EECs has been reviewed extensively by light- and electron-microscopical techniques [31,32,33,34]. All these findings showed that these cells are identifiable by their secretory granules, as reported above. These granules are present in cytoplasmic vesicles; they have a characteristic size and shape for each cell type [35]. Based on these evaluations, it is now widely accepted that ECs are also classified according to their secretory products [32,33,36]. Thus, they were distinguished from other EECs by their pleiomorphic, electron-dense secretory granules [33,37]. Furthermore, Wade and Westfall (1985) showed that ECs extended from the basal lamina of the crypt epithelium to the lumen of the crypt in the duodenum of mice [33] (Figure 8).

As can be seen in Figure 8, ECs showed a tuft of microvilli extending into the lumen of the GI tract, rough endoplasmic reticulum in the perinuclear region, secretory granules concentrated near the nucleus and at the base of the same cell. These granules were different in size and showed oblong, spherical, and biconcave shapes (Figure 9).

A very important finding was that ECs have bundles of nerve fibers below the epithelium basal lamina together with many fenestrated vessels. These considerations suggested that there were synaptic relationships between neuronal fibers and ECs even if, at the time, they had not been confirmed or previously reported in other studies [33]. 

The presence of vessels in proximity to ECs has supported the idea that their secretory products have endocrine functions. Their products were transported by the blood stream to their site of action and finally, when a cell secretes a product, it normally causes a local or paracrine function with potent effects on intestinal tissues [33]. 

These authors did not find granules/secretory products in the apical cellular region. On the contrary, other researchers identified an evident bipolarity of ECs with the presence of secretion products not only in the basal but also in the apical portion of the same cells [34,38]. The presence of granules in the apical portion showed an apical exocrine release of their products; this concept has been supported by the presence of Golgi apparatus in the apical cytoplasm with many microtubules that provide a pathway for the transfer of newly formed secretory products from these organelles to the liminal surface of the ECs. Instead, many of these granules in the basal portion of ECs suggested their endocrine function. 

Further, Gunawardene et al. (2011) showed that ECs are approximately 8 μm in size and triangular or pyramidal in shape [39]. Moreover, these authors showed that ECs have some differences in their morphology: they have basal extensions with secretory vesicles in the colon or rectum. These granules were not seen in the small intestine and the authors, in agreement with the data in Gustafsson et al. (2006), suggested that only these cells are able to communicate with neurons [39]. 

Of particular interest, ECs are involved in GI inflammatory conditions; changes in the content, release, and re-uptake of their secreted products have been studied in inflammatory bowel disease [40]. Their number is increased in clinical and experimental models of inflammatory bowel disease both in affected and non-affected areas of the gut [41].

Nowadays, it is possible to better understand these latter findings as described in Section 2.1 for EECs and as indicated in these references [42,43,44]. 

## 5. Localization of Enterochromaffin Cells

To our knowledge there are few studies evaluating the complex nature of the functions of ECs. The regional and topographical distribution of ECs has been evaluated, in different species, by light- and electron-microscope studies, but this is an important area of research in the present day. A review of the literature and related findings are summarized below.

ECs are scattered along the GI tract with a cell density differing from species to species; this concept continues to be important for defining their role [22,45,46].

In the esophagus, few ECs have been identified in the human epithelium, and their number has been noted to increase in the distal part of the organ [47]. No more findings were found on the localization of enterochromaffin cells. In our opinion, this could be an important new scientific field for research. 

In the stomach, ECs show many variations in species and are numerous in pigs, dogs, rabbits, and monkeys, but rarely present in the gastric corpus and antrum of humans and rats as reported by Bordi et al., (2000) and Hunne et al., (2019) [48,49]. Recently, Koo et al., (2021) confirmed that these cells are present in the gastric antrum of mice; they have swollen bulges containing an accumulation of storage vesicles [26]. These vesicles are approximately 100–150 nm, often smaller than those observed in the GI tract (150–500 nm), [23,50]. 

In the small intestine and in the colon, ECs are evident in humans and other species such as bovines, cats, mice, and guinea pigs. It is known that ECs are mainly present in the colon (43%) and small intestine (22% in the ileum and 20% in the jejunum) in humans, while in rats they are well visible in the cecum (14 × 10^3^/mm^3^ mucosa) and in the ascendent colon (12 × 10^3^/mm^3^ mucosa) with a minor presence in the descendent portion of the same organ (2.5 × 10^3^/mm^3^ mucosa) [22,46,51,52]. 

Sjolund et al. (1983) and Zhang et al., (2012) showed, respectively, the presence of these cells in the rectum of humans and rhesus macaques. To our knowledge, there is no detailed literature about ECs in the rectum of other species [45,53].

## 6. Heterogeneity of Enterochromaffin Cells

It is accepted that there are many subtypes of ECs based on their secreted products, and many of them are colocalized in the same cells [52]. It is important, in fact, to underline that the presence of serotonin is colocalized and mapped with other molecules. To date, these main molecules are melatonin and substance P, although other substances have been described [22,33,38,54].

### 6.1. Serotonin in Enterochromaffin Cells

It is known that about 95% of the body’s serotonin (5-hydroxytryptamine—5-HT) is present in the GI tract, and 90% is packaged in the secretory granules of the basal and the apical portion of ECs [55]. 

Upon stimulation of gut metabolism, ECs promote the release of 5-HT from their granules. Its release is an important mediator stimulating visceral sensation, intestinal motility, and permeability. 

While the release of 5-HT from the apical portion of ECs concerns the phases of gut metabolism reported above, its secretion at the basal portion of the same cells can result in its release into the cardiovascular system. Then, it can be stored in platelets, or it can play its role directly in metabolism likely to neuroendocrine substance [2,22,56,57,58,59]. 5-HT present in the blood stream is considered to act negatively on osteoclast proliferation and positively on hepatic regeneration [60].

#### Serotonin–Gut Microbiota Interaction

It is well known that the interaction between 5-HT and the gut is highly complex and bidirectional. This bidirectional link demonstrates a mechanism by which 5-HT, released from ECs, can modulate the microbiota species, glucose, and lipid metabolism. Interestingly, Jones et al. (2020) indicated that the apical release of 5-HT positively activates the microbiota and vice versa inducing the proliferation of ECs [12]. Instead, the release of basal 5-HT can activate vagal afferent fibers inducing intestinal motility and, through gut-brain axis signaling promote gastric emptying. Moreover, when 5-HT is released in the blood stream can stimulate glucose and lipid metabolism (Figure 10).

Another interesting point is that ECs are electrically excitable like sensory primary cells because they have voltage-gated sodium (Na^+^) and calcium (Ca^2+^) channels; moreover, they use specific receptors and signal networks to recognize important signals [2]. These receptors are the transient receptor potential A1 (TRPA1) ion channel, which is somatosensory for exogenous dietary irritants or endogenous inflammatory agent receptors; the microbial metabolite sensor olfactory receptor 558 (Olf-558); and the stress response-related signaling cascade mediated by the α2A adrenoreceptor (Adrα2A)-TRPC4 channel [61]. All these receptors stimulate sodium and calcium channels involving the release of 5-HT from ECs through neuronal vagal fibers. 

Currently, it is known that these cells show plasticity as EECs and respond to several signaling from different species or in physiological or pathophysiological states. Interestingly, in physiology the release of 5-HT from ECs is mechanically activated through food bolus-induced pressure towards the intestinal wall, which results in direct activation of the mucosa of ECs [58]. Another interesting point concerns duodenal ECs; they secrete 5-HT into the mesenteric lymph in response to glucose and, for this reason, the sodium-glucose transporter-1 and 3, abundantly expressed in small intestine, have been investigated as glucose sensors of these cells [61]. The intestinal absorption of glucose is mainly carried out over the length of the small intestine, on the contrary, in physiological conditions, only a limited quote of glucose is taken up by colon. Moreover, the presence of sugar transporters such as GLUT and Sglt, also in the colon, indicates a capacity for glucose absorption in this region. Data confirm that the capacity of ECs to sense the nutrient content of the GUT depend upon their location through the overall length [62]. Moreover, other results underlines that it is the Sglt-1 mediated pathway the one that mainly influence 5-HT release from ECs, confirming that also in colon the release of 5-HT is consequent to luminal glucose-stimulus [62]. The glucose specificity of this organ supports the possibility of an uncharacterized glucose-sensing mechanism. So, since dietary sugars may differently stimulate 5-HT release in different way, ECs may be heterogeneous in terms of nutrient-sensing capacity based on their location in the GI tract [62,63,64,65]. 

In addition, ECs are also sensitive to endogenous molecules, including some neurotransmitters [66]. In this regard, norepinephrine is a potent bacterial stimulus and its stimulation by ECs could be protective by inducing activation of GI motility for expelling microbes and harmful chemicals [2]. However, persistent stimulation of afferent nerve fibers and ECs could elicit visceral hypersensitivity. Then, as reported by Kim and colleagues (2004) and Westlund et al. (2014), polymorphism or deletion of Adrα2A-TRPC4 genes could determine visceral pain syndromes [67,68]. 

Based on these findings, ECs can modulate mechanosensory function, but they are themselves mechanosensitive even if their contribution has not yet been well defined [69]. 

Many studies suggested that ECs signaling diversity and plasticity are also evident at neuronal circuits. 5-HT interacts with 5HT_3_-receptors that are expressed on nerve fibers; these fibers act on numerous EECs to induce several signals (peptide hormones, serotonin) to regulate nerve activity or modulate other neuronal activity [5,70]. Furthermore, 5-HT can act on enteric neurons, immune cells or be taken up by platelets as reported above. 

It has also been demonstrated that ECs through their production of 5-HT, play a role in the control of insulin resistance, hepatic gluconeogenesis, thermogenesis, and obesity [65,71,72]. 

The above findings suggest a very useful role of 5-HT release from ECs, but it has been found that the proliferation of these cells and high presence of 5-HT in their vesicles are involved in visceral sensitization and hyperalgesia in irritable bowel syndrome [43]. If some receptors on duodenal mucosa ECs such as free fatty acid receptors are activated, ECs protect the small intestine mucosa regulating 5-HT biosynthesis and release, but if the stimulation of the receptors is excessive can induce the abnormal release of 5-HT. Excessive release of 5-HT can cause mucosal damage and a reduction in blood flow [73]. In this case, ECs are sentinels for sensing the variations in gut mechanisms and their functions are not yet well known; Xu et al., (2021) and Bi et al., (2021) demonstrated that knockout mice for 5-HT transporters showed visceral sensitization, GI disorders and an increased the relative proportion of ECs and colon 5-HT concentration [43,74]. This mechanism has been explained by suggesting that the release of 5-HT from ECs is related to adenosine triphosphate; this can activate the pain network to the brain through afferent nerve fibers and enteric nerve reflex [74]. 

Of particular interest, the three receptors for 5-HT are expressed in several immune cells of humans and rodents [75,76]. It is known that the recruitment of immune cells (i.e., dendritic cells, eosinophils, and mast cells) in the inflammation site is stressed by 5-HT during acute inflammation (Figure 11).

Moreover, these authors suggested that ECs and 5-HT secretion are linked to the etiology of several diseases such as inflammatory bowel disease, celiac disease, and neuroendocrine tumors. 5-HT appears to be a critical molecule in maintaining homeostasis in the gut and the disruption of the balance signaling induces the development of the pathological conditions as mentioned above (Figure 12).

### 6.2. Melatonin and Enterochromaffin Cells

Alongside 5-HT, ECs are the prime source of melatonin in the GI tract; originally discovered in the pineal gland, it is now known that melatonin is also produced in the GI tract from the stomach to the colon. Russian scientists were the first to demonstrate the active melatonin synthesis in human gut ECs [77]. This discovery has expanded the importance of extra-pineal melatonin sources in mammals. 

It is known that ECs produce several hundred times more melatonin than is produced by the pineal gland [78]. 

Melatonin has been studied in the GI tract using several approaches; it is biosynthesized from serotonin through the acetylserotonin O-methyltransferase and arylalkylamine-N-acetyltransferase enzymes [79]. These rate-limiting enzymes for melatonin synthesis have been found in ECs, which confirms the local production of this indolamine [78]. The presence of these enzymes in ECs suggests that their biosynthesis is the same as in pinealocytes [78]. The biosynthesis is illustrated in Figure 13. Briefly ECs take up L-tryptophan from the circulation and by tryptophan-5-hydrohylase or, monooxygenase is converted into 5-hydroxy-tryptophan (5-HTP). Then, 5-HTP is transformed into 5-HT by aromatic l-amino acid decarboxylase [80,81].

#### Role of Melatonin in Gastrointestinal Tract

Melatonin is not uniformly present in the GI tract, and its concentration varies by GI portions and among species [82]. It has been detected mainly in the nuclear fraction and in minor amount in microsomal, mitochondrial, and cytosolic fractions [83,84].

The GI tract is the main site exposed to the oxidative stress and inflammation due to the presence of microbes and pathological bacteria. Thus, it needs strong protection; melatonin synthesized by ECs is a potent antioxidant molecule and it is also an effective inflammatory agent [78]. It scavenges a broad spectrum of reactive oxygen species (ROS) by upregulating several antioxidant enzymes and downregulating pro-oxidant enzymes [85]. In addition, the interaction with ROS induces the production of many molecules with several antioxidant capacities. This mechanism has been defined as an antioxidant cascade reaction, so that one melatonin molecule can destroy up to 10 ROS. This capacity makes melatonin highly protective, even at low concentrations, in defending the body from oxidative stress [86].

Melatonin effects in the GI tract are mediated by membrane receptors (MT1 and MT2); for example, in the stomach it has positive effects on the gastric mucosa against stress and ischemia and quickly induces ulcer healing [64,87]. The scientists demonstrated that both MT1 and MT2 receptors are present on the mitochondrial membrane of gastric endothelial cells; in particular, MT1 is expressed in the mitochondrial membrane while MT2 is also present in the nucleus [87,88]. They supposed that, in the gastric endothelial cells, the effects of melatonin via mitochondria increases its protection in the GI tract.

It is known that melatonin inhibits cyclic adenosine monophosphate (cAMP) production which can induce hydrochloric acid secretion by the parietal cells, so melatonin suppresses hydrochloric acid production [78]. Moreover, melatonin levels are linked to 5-HT concentrations since this latter is its common biosynthetic precursor [79].

It has previously been demonstrated that the release of melatonin from the GI tract is linked to food intake. No food induces an increase of melatonin in plasma and the GI tract, so the authors suggested that melatonin can act as an autocrine and paracrine indolamine involving not only the GI epithelium but also smooth muscle of the GI [89]. This point has been stressed in studies by Huether (1993) studies using pinealectomized animals. These animals showed high levels of melatonin in the blood flow suggesting that, under some situations, the gut may deliver melatonin into the cardiovascular system [90]

Further studies indicated that melatonin is protective of GI microbiota by inducing the increase of beneficial bacteria and reducing negative bacteria in different pathologies [91,92]. This focus is an important area of research that is not well explored to date.

### 6.3. Substance P and Enterochromaffin Cells

Substance P (SP) was originally identified by von Euler and Gaddum (1931) in the central nervous system [93]. Later, SP was localized in nerves and in the mucosa of the GI tract of many species [94,95]. Then, Heitz and colleagues (1976) found SP in ECs of rabbit bile duct and human small intestine (duodenum), although in a lesser amount [96].

SP is a primary sensory neurotransmitter like somatostatin and vasoactive intestinal peptide (VIP); high levels of this peptide have also been identified in cerebral grey matter and the autonomic system [96,97,98,99]. Furthermore, Alumets et al. (1977) visualized, using a combination of ultrastructural and immunohistochemical techniques, a moderate and strong immunoreactivity for SP in certain intestinal carcinoid tumors that derive from ECs [99]. This positivity has been observed in the cytoplasmic granules of tumoral cells together with 5-HT positivity. The authors concluded that both 5-HT and SP are produced, stored, and secreted by ECs.

SP has been noted to have a large variety of functions; these include motility, fluid and electrolyte secretion, blood flow, and regulation of the immunoinflammatory response. Furthermore, SP has proinflammatory effects in immune and epithelial cells and participates in inflammatory diseases of the respiratory, gastrointestinal, and musculoskeletal systems and exerts its biological activity on target cells by interacting mainly with the neurokinin-1 receptor [100,101,102,103].

Moreover, Machida et al. (2017) and Obara et al., 2018 showed that methotrexate and cisplatin, respectively, caused a significant increase in the number of ECs containing substance P after 24 h after their administration; these drugs induce emesis and the authors suggested that these results are very important for understanding the GI damage induced by chemotherapy with these drugs [104,105].

We can speculate that many functions of SP as well as those of 5-HT in ECs of the GI tract and throughout the body are still unknown.

## 7. Enterochromaffin Cells and Diet in Pathology

EECs and ECs directly integrate signals of nutrients in the gut lumen with biological responses inducing the production of several substances as reported above.

Given the role of ECs in the maintenance of digestive tract homeostasis and their alterations in metabolic syndromes [106,107,108,109], we describe the recent developments concerning the relevance of diet on these cells in health and in illness.

It is known that the change in the distribution and in the number of ECs can depend on the diet type and the gut microbiota [110,111]. These authors studied the health status of pigs fed with a rye-type and hybrid rye-type grains, and they showed that neither of the rye grains modified the homeostasis and the somatostatin production by neuroendocrine cells. These results are important in a wider strategy for developing a new nutritional plan for pigs [110]. Successively, the same authors evaluated the number of the ECs in the small intestine of suckling piglets stimulated with red kidney bean lectin [111]. They showed an increased number of these cells producing serotonin in the duodenum and they suggested that this treatment is stimulant of the gut maturation in suckling mammals.

ECs act as important luminal sensors within gastrointestinal tract and can sense several nutrient stimuli and secrete 5-HT in response to luminal sugars [62,112]. In humans, the sensing of glucose is altered with obesity and the researchers assessed that the mice feed with high fat (HFD) diet began obese [112,113,114]. Martin et al. (2020) showed that HFD-feeding alters the sensing of ECs in a region-dependent manner involving duodenum but not colon. In the duodenum, ECs change their functions in relation to minor perception to nutrients sensitivity; this diet increases only the number of ECs in the colon, which is correlated to increase in circulating 5-HT. The findings suggested that the increase of ECs in the colon is due to 5-HT release like observed in obese humans and indicated a strong correlation among ECs and both 5-HT concentration and fasting blood glucose. They highlighted an important mechanism by which 5-HT is involved in metabolic disease progression. The mechanism is not yet well known, but it can be due to the changes in the gut microbiota, which determines a metabolic phenotype as observed through the transfer of the same from lean and obese individuals [115,116]. In this regard, Beyec et al., (2014) demonstrated that the leptin is upregulated in obese patients and its increase is due to the 5-HT overexpression from ECs occurring before the onset of obesity and expression of fat mass. This finding is relevant in the pathophysiology of obesity [117].

Inflammatory bowel disease is a chronic intestinal inflammatory disease affecting patients’ quality of life [118]. The patients with this disorder report that food aggravates their problems. Accumulating evidence has demonstrated that the interactions between specific types of foodstuffs and ECS result in change in the cell density. These cells decreased but their number is restored after an individual dietary guidance [119].

Considering all these findings, we would like to emphasize and stress the different capabilities of ECs and the complexity in understanding how their release of certain substances, such as 5-HT, may locally impact local the gastrointestinal tract and systemic processes [120].

## 8. Conclusions and Perspectives

It is appropriate to conclude this overview of the main findings on EECs with a summary of issues reported in the previous paragraphs.

As the major cell type of ECs in the GI tract, our goal was to specifically consider these cells. They are distributed in the GI tract with different cell density from species to species.

It is becoming clear that ECs play a major role in managing all the signals that come from the external and internal body to preserve intestinal immune homeostasis under steady-state conditions. Furthermore, ECs increase in number during diseases (inflammatory bowel diseases, carcinoid tumors, hyperalgesia), and during chemotherapy and irradiation (vomiting and diarrhea effects), but surprisingly the major contribution of these cells to pathological conditions is linked to the alterations in secretion products. These changes concern the higher amount of these products and their abnormal biological signaling; therefore, the secretion, availability and re-uptake mechanisms of these substances could contribute to the etiology of diseases.

5-HT, melatonin, and SP play an important role in the functions of ECs in the GI tract even if there are other secretion products that are secreted in smaller amounts. The mechanisms by which 5-HT, melatonin, and SP act are partially unknown and not well investigated. Thus, better understanding how ECs and their secretions participate in regulating biological functions of the body associated also to diet could be crucial to finding new therapeutic targets in several disorders. An understanding of these processes could help with determining dietary strategies that modulate the secretion of these peptides. Such diets could be used as a protective measure against or as a part of a treatment for metabolic diseases.

## Figures and Tables

**Figure 1 ijms-23-03758-f001:**
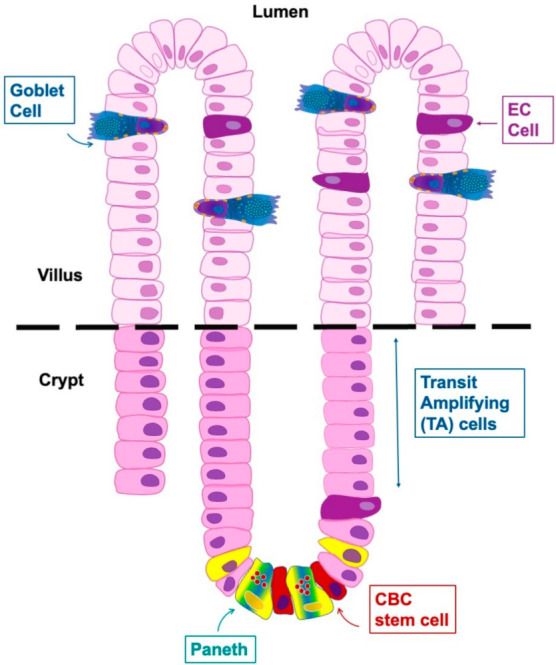
Schematic representation of the differentiation and functional anatomy of the crypt-villus unit. EC: enterochromaffin cell; CBC: rapidly cycling crypt base columnar cells; TA: proliferative transit-amplifying zone. Modified from Li et al., 2011 [10].

**Figure 2 ijms-23-03758-f002:**
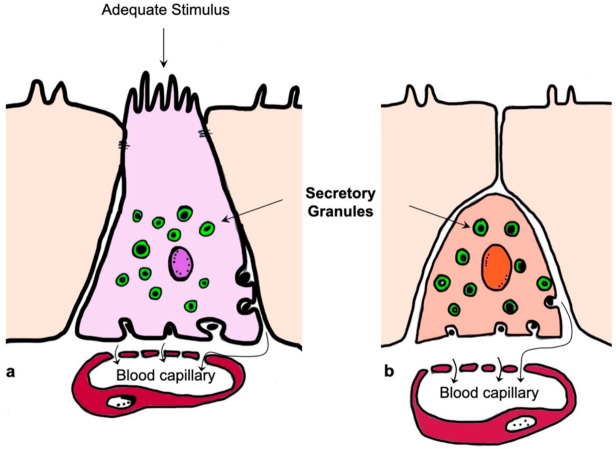
Schematic representation of the enteroendocrine cells; open (**a**) and closed (**b**) type. Modified from Sundler et al., 1989 [14].

**Figure 3 ijms-23-03758-f003:**
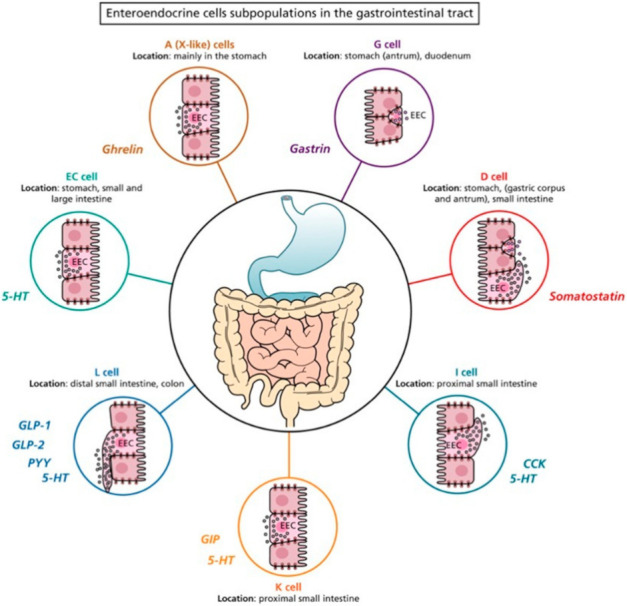
Schematic representation of the enteroendocrine cell subpopulation in the gastrointestinal (GI) tract. EC: enterochromaffin cell; 5-HT: 5-hydroxytryptamine; CCK: cholecystokinin; GLP-1: glucagon like peptide-1; GLP-2: glucagon like peptide-2; PYY: peptide YY; GIP: gastric inhibitory peptide-1; dark red circle indicates the A cells secreting ghrelin in the stomach (corpus); purple circle indicates the G cells secreting gastrin in the stomach (antrum); red circle indicates the D cells secreting somatostatin in the stomach (pylorum); dark blue circle indicates the I cells and orange circle indicates the K cells secreting CPK and GIP in the small intestine; light blue circle indicates the L cells secreting GLP-1, GLP-2 and PYY, widely distributed in small intestine and in colon. The Figure is obtained from Latorre et al., 2016. Reprinted under the terms of the Creative Commons CC BY license [8].

**Figure 4 ijms-23-03758-f004:**
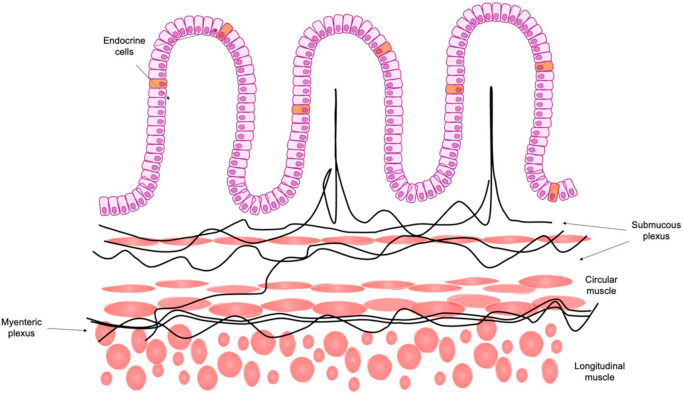
Schematic representation of the enteroendocrine cells, in particular, a focus on their anatomic relationship with the enteric nervous system. Figure modified from El-Salhy M., 2020 [17].

**Figure 5 ijms-23-03758-f005:**
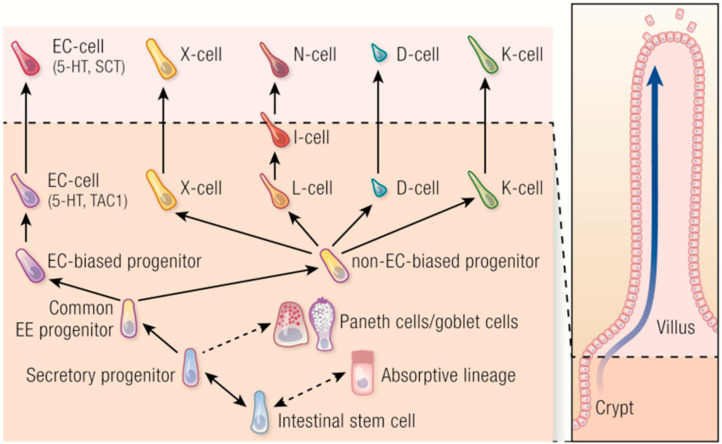
Schematic representation of the different cells that could differentiate from the intestinal stem cells. They could present an absorptive pattern or a secretory activity, including enteroendocrine cells, Paneth cells, and goblet cells. EC: enterochromaffin cell; black arrows and black dotted arrows represent the differentiation process from staminal to intestinal specific cells. The Figure is obtained from Beumer et al., 2020. Reprinted under the terms of the Creative Commons CC BY license [3].

**Figure 6 ijms-23-03758-f006:**
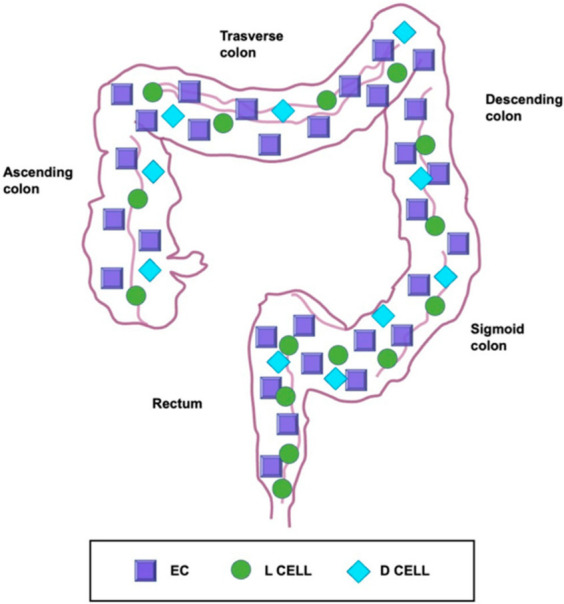
Schematic representation of the different distribution of enteroendocrine cell subtypes. They are the most abundant enteroendocrine cell subtype of the colon and rectum. D cells are scattered evenly throughout the GI tract, even if they are uncommon. The frequency of L-cells, instead, increases from proximal to distal being most concentrated in the rectum. EC: enterochromaffin cell. Modified from Gunawardene et al., 2011 [23].

**Figure 7 ijms-23-03758-f007:**
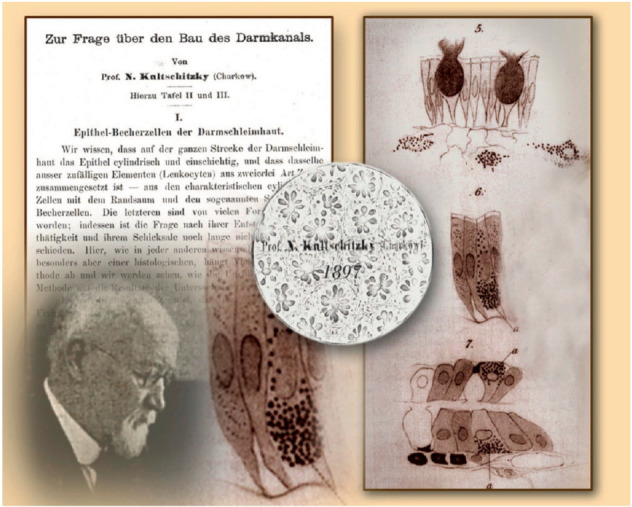
First description of a «unique» cell of the intestinal epithelium (right), made by Nikolai Kulchitsky (bottom left) in 1897. His work Zur Frage über den Bau des Darmkanals was published in Arch Mikr Anat 1897. The Figure is obtained from Drozdov et al., (2008). Reprinted by permission from Springer Nature Customer Service Centre GmbH: Springer Nature, Cell and Tissue Research (License Number 5232431160386, 19 January 2022) [24].

**Figure 8 ijms-23-03758-f008:**
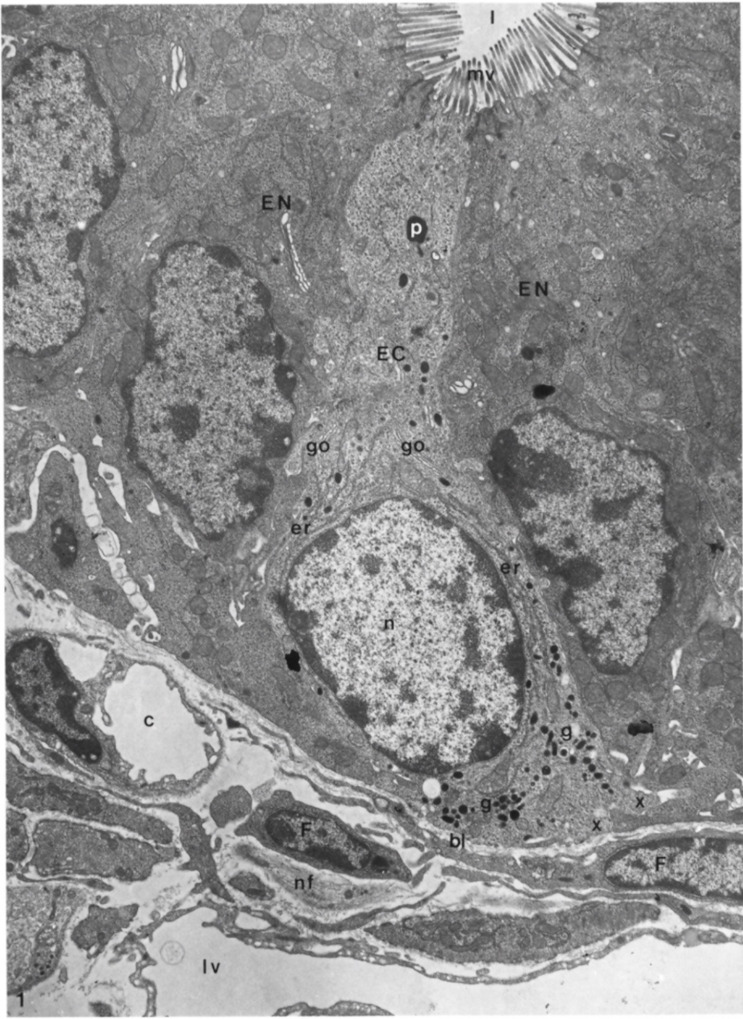
Longitudinal section of an enterochromaffin cell from the basal lamina to the lumen of a duodenal crypt, in which it is possible to recognize a fenestrated capillary, fibroblast, lymph vessel, and nerve fibers. Moreover, it is also possible to highlight the presence of rough endoplasmic reticulum, secretory granules, Golgi complexes, microvilli, phagosome, and basal cytoplasmic extensions of the enterochromaffin cell. EC: enterochromaffin cell; bl: basal lamina; l: lumen; c: capillary; nf: nerve fibers; lv: lymph vessel; F: fibroblasts; EN: enterocytes granules; g: secretory granules; go: Golgi complexes; mv: microvilli; p: phagosome; x: basal cytoplasmic extension. X 9.600. The Figure is obtained from Wade et al., (1985). Reprinted by permission from Springer Nature Customer Service Centre GmbH: Springer Nature, Cell and Tissue Research (License Number 5232430326152, 19 January 2022) [33].

**Figure 9 ijms-23-03758-f009:**
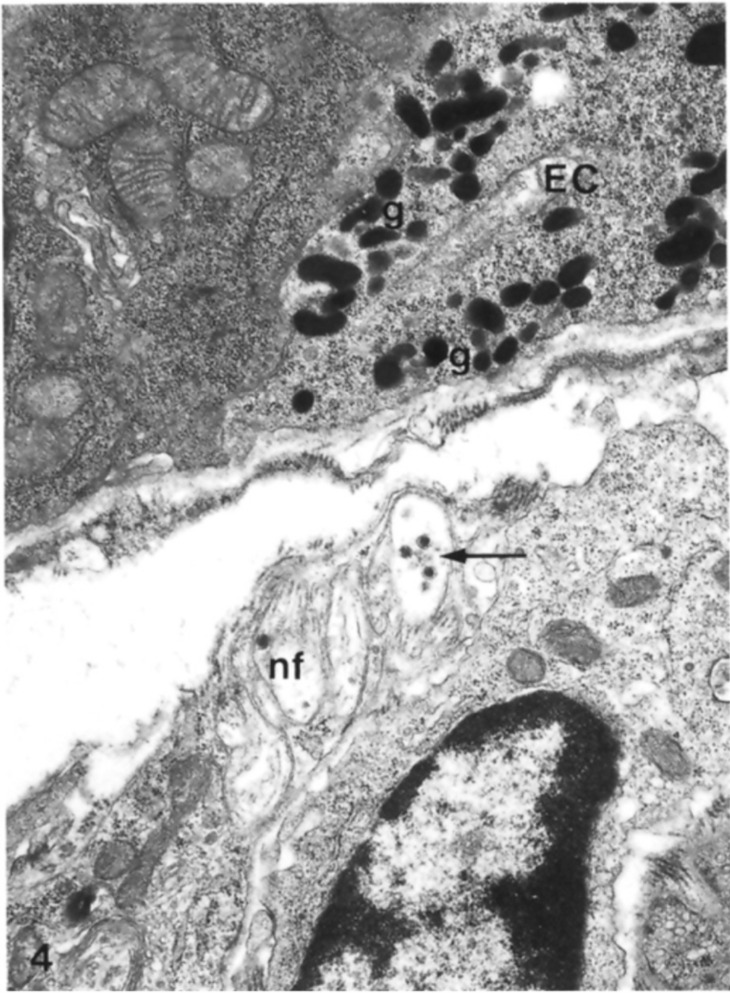
Base of an enterochromaffin cell fixed in glutaraldehyde and osmium tetroxide in which it is possible to see the presence of secretory granules containing clear and dense-cored vesicles, close to nonmyelinated nerve fibers. EC: enterochromaffin cell; nf: nonmyelinated nerve fibers; g: secretory granules; black arrows indicate dense-cored vesicles. X25.500. The Figure is obtained from Wade and Westfall, (1985). Reprinted by permission from Springer Nature Customer Service Centre GmbH: Springer Nature, Cell and Tissue Research (License Number 5232430326152, 19 January 2022) [33].

**Figure 10 ijms-23-03758-f010:**
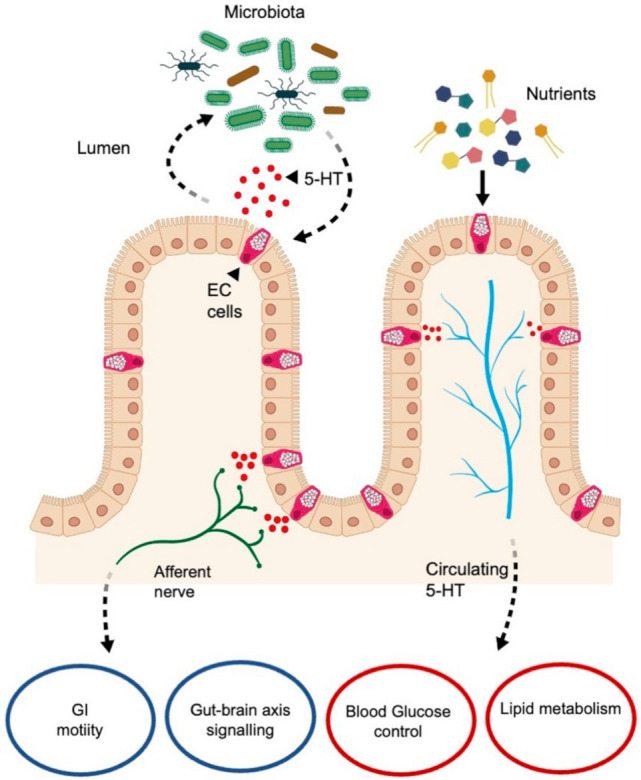
Schematic representation of the bidirectional relationship among ECs, enteric nervous system, 5-HT, nutrients, and microbiota. 5-HT released from EC cells can modulate the microbiota species, but moreover, once released in the blood stream, it could stimulate glucose and lipid metabolism. Basal 5-HT’s release can also activate vagal afferent fibers inducing intestinal motility and modulating the brain–gut axis signaling. EC cell: enterochromaffin cell; 5-HT: 5-hydroxytryptamine; GI: gastrointestinal. Black arrows and black dotted arrows represent the relationship between intestinal mucosa, the nervous system, the circulatory system and the lumen’s content. Modified from Jones et al., 2020 [12].

**Figure 11 ijms-23-03758-f011:**
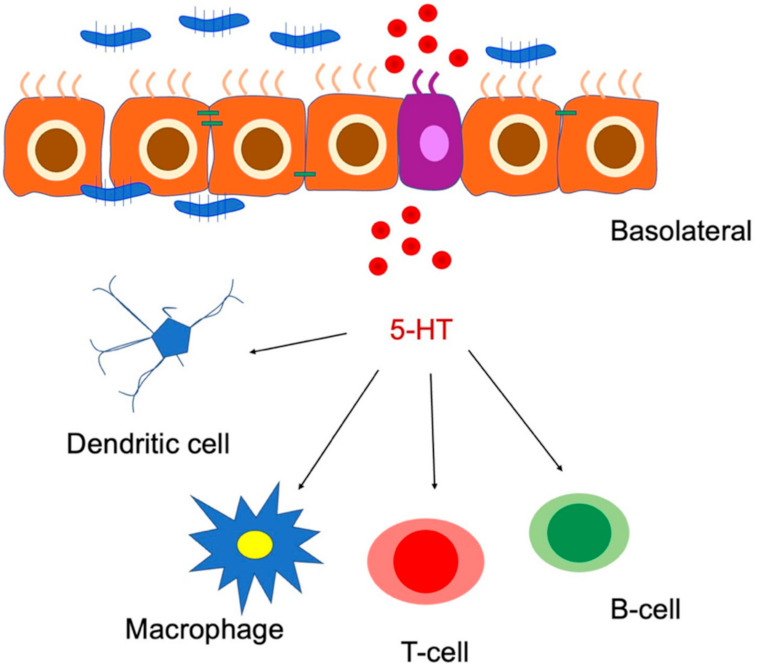
Schematic representation of the role of 5-HT in immune cell activation and gut inflammation. 5-HT: 5-hydroxytryptamine. Modified from Banskota et al. (2019) [76].

**Figure 12 ijms-23-03758-f012:**
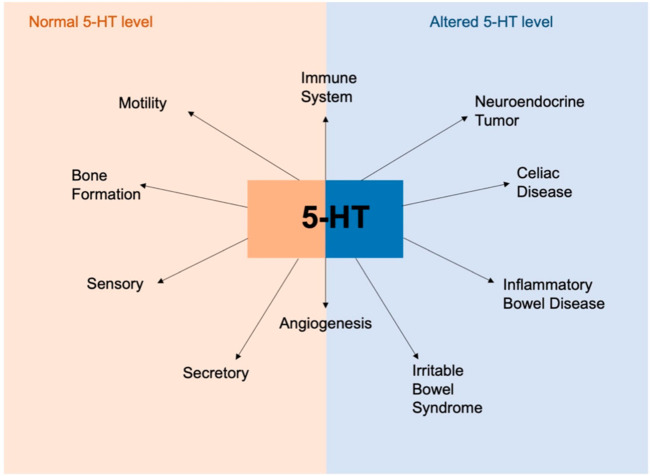
Overview of the involvement of 5-HT in physiological and pathophysiological conditions. 5-HT plays an important role in gut physiology and in maintaining intestinal homeostasis. 5-HT is also important in the regulation of immune response and angiogenesis. Alteration in 5-HT production resulting in dysregulation of 5-HT signaling is associated with various disorders like inflammatory bowel diseases, celiac disease and neuroendocrine disorders. 5-HT: 5-hydroxytryptamine. Modified from Banskota et al., 2019 [76].

**Figure 13 ijms-23-03758-f013:**
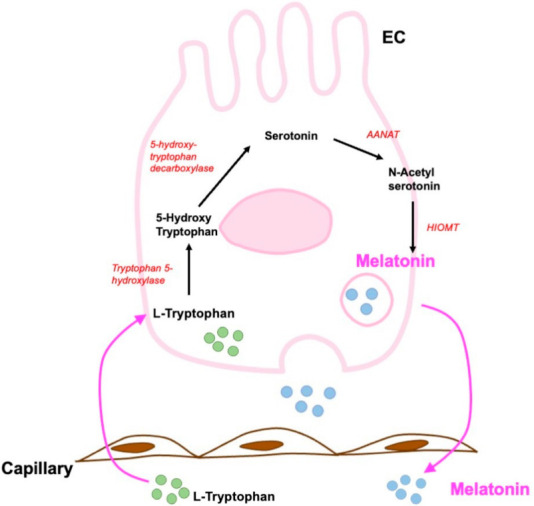
Schematic description of the biosynthetic pathway of melatonin in enterochromaffin cells of mammals. ANAAT: arylalkylamine-N-acetyl transferase; HIOMT: hydroxindole-O-methyltransferase; EC: enterochromaffin cell; black arrows indicate the reactions. Modified from Pal et al., 2019 [78].

**Table 1 ijms-23-03758-t001:** Enteroendocrine cells (EECs) and their secreted products in the gastrointestinal (GI) tract. (5-HT: 5-hydroxytryptamine; CCK: cholecystokinin; GLP-1: glucagon like peptide-1; GLP-2: glucagon like peptide-2; PYY: peptide YY; GIP: gastric inhibitory peptide-1).

EECs	Location	Hormones
A (X-like) cells	Stomach (corpus)	Ghrelin
D cells	Stomach (pylorum) Proximal segment of the small intestine	Somatostatin
**EC cells**	Stomach (pylorum), small and large intestine	5-HT
G cells	Stomach (antrum), duodenum	Gastrin
I cells	Small intestine Distal portion small intestine	CCK 5-HT
L cells	Distal portion of small intestine Colon (mainly in the proximal portion) Large intestine	GLP-1, GLP-2, PYY 5-HT
K cells	Small intestine	GIP 5-HT

## Data Availability

Not applicable.

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
