# Peer review of "A Focus on Enterochromaffin Cells among the Enteroendocrine Cells: Localization, Morphology, and Role"

_ijms, 2022, doi:10.3390/ijms23073758_

Round 1
Reviewer 1 Report
The reviewed article addresses the role of EEC cells in the gut.
It is generally well written, and my comments aimed to increase the scientific soundness and clarity of it.
My minor concerns:
- The topic of this article is not new and has been presented several times before. There are many more comprehensive articles dealing with the same subject. To make this article more interesting to a reader I suggest the authors to precisely and clearly define the target question of this review.
- The authors did not mention that at least some of dietary components were found to change the proportion of EEC in the gut. For example both stimulation with red kidney bean lectin (Zacharko-Siembida et al. "Red kidney bean (Phaseolus vulgaris) lectin stimulation increases the number of enterochromaffin cells in the small intestine of suckling piglets" Journal of Veterinary Research, vol.58, no.2, 2014, pp.289-294. https://doi.org/10.2478/bvip-2014-0044) as well as feeding with hybrid rye grain (Zacharko-Siembida et al. “Expression of serotonin, somatostatin, and glucagon-like peptide 1 (GLP1) in the intestinal neuroendocrine cells of pigs fed with population rye type and hybrid rye type grains” Med. Wet. nr 75 (10), 2019, ss. 593-598, DOI:10.21521/mw.6251) increased the proportions of EEC in the pig small intestine. Both works and similar must be included and discussed in this review.
- Line 91 – Why "Gastrointestinal" is written in capital ? The pylorum is also a part of the stomach. So more appropriate would be to write: “stomach (corpus)”, stomach (antrum) and “stomach (pylrorus)”.
- Figure 4 – the location of submucosal plexus is wrong. The authors in fact, indicated the position of myenteric plexus only. The submucosal plexus is generally placed between submucosa and mucosa layers (in pigs and humans it is further divided into outer and inner submucosal plexuses).
Author Response
Reviewer 1 Comments |
Changes made by the authors in response to the comments (in blu in the text) |
The topic of this article is not new and has been presented several times before. There are many more comprehensive articles dealing with the same subject. To make this article more interesting to a reader I suggest the authors to precisely and clearly define the target question of this review. |
We thank the Referee for the suggestion. We have better defined our aim and objectives in the first paragraph of the review (pages 1 and 2) and in the conclusive paragraph (page 18). |
The authors did not mention that at least some of dietary components were found to change the proportion of EEC in the gut. For example both stimulation with red kidney bean lectin (Zacharko-Siembida et al. "Red kidney bean (Phaseolus vulgaris) lectin stimulation increases the number of enterochromaffin cells in the small intestine of suckling piglets" Journal of Veterinary Research, vol.58, no.2, 2014, pp.289-294. https://doi.org/10.2478/bvip-2014-0044) as well as feeding with hybrid rye grain (Zacharko-Siembida et al. “Expression of serotonin, somatostatin, and glucagon-like peptide 1 (GLP1) in the intestinal neuroendocrine cells of pigs fed with population rye type and hybrid rye type grains” Med. Wet. nr 75 (10), 2019, ss. 593-598, DOI:10.21521/mw.6251) increased the proportions of EEC in the pig small intestine. Both works and similar must be included and discussed in this review. |
We thank the Referee for the precious consideration. We have introduced an additional paragraph (7. Enterochromaffin cells and diet in pathology, page 17 and 18) for better depicting this additional aspect in order to give a more complete overview of the state of the art around EC. Moreover, we have also added some considerations in the conclusive part (page 18). |
Line 91 – Why "Gastrointestinal" is written in capital? The pylorum is also a part of the stomach. So more appropriate would be to write: “stomach (corpus)”, stomach (antrum) and “stomach (pylrorus)”. |
We thank the Referee for the suggestion: we corrected the capital letter and we uniformed in the text and in Table 1 the description of the different stomach’s regions. |
Figure 4 – the location of submucosal plexus is wrong. The authors in fact, indicated the position of myenteric plexus only. The submucosal plexus is generally placed between submucosa and mucosa layers (in pigs and humans it is further divided into outer and inner submucosal plexuses). |
We thank a lot the Referee for the annotation: in the first original version our Figure 4 was obtained from the Figure reported in another article, but, as the Referee pointed out, it is not completely representative of what we want to explain. For this reason, we decided to replace the image with another figure that we have remodeled from Figure 1 of El-Salhy M.; 2020. |

Reviewer 2 Report
This is a nice review to summarize the current knowledge on enterochromaffin cells (EC).; however, the authors mainly depend on the previous reviews and not on the original papers. In addition, there are some mistypes between EC, EEC, and ECC?, which induce the misleading of the readers. I am pointing some related points in order to recommend the authors to confirm each descriptions and references.
L50, 62, 68, 77, 80
ECCs? EECs ?
L298
ref [39]
It is better to refer more recent paper.
L309
ref [55]
Is it correct ?
L309-L316
The difference between duodenal and colon ECs is not clear.
L343
Xu et al, 2021; Bi et al., (2021)
The description is correct ?
L349 and other places
5-HT3? 5-HT?
L401-L407
It is reasonable that melatonin receptors work as membrane protein in the cell surface. If they are locate to the mitochondrial or nuclear membrane, how the melatonin reach to the receptors?
L449
EECs >ECs?
L451
ECs >.EECs ?
Author Response
Reviewer’s 2 questions |
Changes made by the authors in response to the comments (in green in the text) |
This is a nice review to summarize the current knowledge on enterochromaffin cells (EC).; however, the authors mainly depend on the previous reviews and not on the original papers. In addition, there are some mistypes between EC, EEC, and ECC?, which induce the misleading of the readers. I am pointing some related points in order to recommend the authors to confirm each descriptions and references.
L50, 62, 68, 77, 80
ECCs? EECs? |
We thank the Referee for the suggestion: we corrected the typing mistakes through the text aligning the abbreviations as follows: - EEC: enteroendocrine cell - EECs: enteroendocrine cells - EC: enterochromaffin cell - ECs: enterochromaffin cells |
L298 ref [39] It is better to refer more recent paper. |
We thank a lot the Referee for the annotation: we checked the references, removing the present one and adding a more recent article (Bellono et al., 2017). |
L309 ref [55] Is it correct? |
We thank a lot the Referee for the annotation: we checked the reference 55 and it is correct. |
L309-L316 The difference between duodenal and colon ECs is not clear. |
We thank the Referee for the suggestion: we added a few sentences to better specify the difference in the role of glucose concentration and glucose sensing for EC’s activity in duodenum and colon (page 13, from Martin et al., 2017). |
L343 Xu et al, 2021; Bi et al., (2021) The description is correct? |
We thank the Referee for the suggestion: we checked the references (Xu et al., 2021) and we have slightly reshaped the sentence (page 13). |
L349 and other places 5-HT3? 5-HT? |
We thank a lot the Referee for the annotation: we checked in some references (Bi et al., 2021) and we correct the typing mistake indicating the 5-HT receptor 3 in the correct way (page 14). |
L401-L407 It is reasonable that melatonin receptors work as membrane protein in the cell surface. If they are locate to the mitochondrial or nuclear membrane, how the melatonin reach to the receptors? |
We thank a lot the Referee for the annotation: melatonin is a pleiotropic molecule. First, once released in the bloodstream, MT act as a hormone, so, as any other hormones it is possible to recognize specific MT-receptors, in particular, MT1 and MT2, membrane receptors that are the main responsible for its ‘‘chronobiotic’’ effects (Dawson and Armstrong, 1996; Hardeland et al., 2011). Secondly, it can exert non-mediate mechanisms, without the interactions with its cellular receptors; in this case MT works directly on other molecules as happens with its antioxidant action. In the end, considering its molecular features, MT can act also through the modulation of gene expression. |
L449 EECs >ECs? L451 ECs >.EECs ? |
We thank the Referee for the annotation: also in this case we want to refer to enteroendocrine cells (EECs) and to enterochromaffin cells (ECs). We corrected the typing mistakes through the text aligning the abbreviations as follows: - EEC: enteroendocrine cell - EECs: enteroendocrine cells - EC: enterochromaffin cell - ECs: enterochromaffin cells |
